# Identifying sea breezes from atmospheric model output (sea\_breeze v1.1)

Andrew Brown<sup>1,2</sup>, Claire Vincent<sup>1,2</sup>, and Ewan Short<sup>2</sup>

<sup>1</sup>ARC Centre of Excellence for 21st Century Weather, The University of Melbourne, Australia

<sup>2</sup>School of Geography, Earth and Atmospheric Sciences, The University of Melbourne, Australia

**Correspondence:** Andrew Brown (a.brown1@unimelb.edu.au)

Abstract. The sea breeze is a mesoscale atmospheric circulation that has implications for human activity and wind energy availability in coastal areas. Sea breezes have been studied in many regions throughout the world, with analyses usually identifying them at individual coastal sites based on local characteristics. Therefore, there is currently a lack of robust and generalizable identification methods, resulting in difficulties analyzing sea breeze characteristics over large regions. Here, software is developed that applies three, physically-based diagnostics for sea breeze identification to atmospheric model datasets. The diagnostics are tested across the coastline of Australia over a 6-month period. These diagnostics identify sea breezes based on either a front or circulation, with additional filtering applied to each diagnostic to reduce mis-classifications. The diagnostics are tested on four different model datasets, ranging from 2.2 km to around 25 km horizontal grid spacing, to explore the impact of spatial resolution on sea breeze identification methods. Based on a range of individual cases, as well as statistics of sea breeze occurrences, we suggest that a method based on moisture frontogenesis may potentially be suitable for sea breeze identification from model data. However, results for individual sea breeze cases indicate that there are difficulties associated with separating the sea breeze from other coastal fronts and circulations. These results have applications for quantifying the effect of sea breezes on human activities, such as for coastal wind energy and the modulation of the urban heat island.

# 1 Introduction

The sea breeze is a mesoscale atmospheric circulation relevant for many aspects of society, including human health (Zhou et al., 2021), transport of pollution (Lawrence and Lelieveld, 2010), and wind energy resources (Steele et al., 2015). The sea breeze is driven by differential surface heating over land and sea during the day, and can be characterized as a thermally-direct circulation with onshore flow and ascent over the land and offshore flow with descent over the sea. However, the sea breeze forms against the complexity of the prevailing background winds (Qian et al., 2009), and can also be defined via the linear gravity-wave like solutions of Rotunno (1983). There can also sometimes be a time lag between the peak in the forcing, and the peak in the wind response, due to the confounding effects of rotation and friction (Du and Rotunno, 2015). The formation of the sea breeze circulation is also associated with a front over the land between maritime and terrestrial air. The sea breeze front can propagate hundreds of kilometers inland before dissipating through frictional processes (Bao et al., 2023). With the

50

passage of this front, there is typically a local decrease in temperature and an increase in moisture, as well as an increase in the onshore component of the wind.

Previous studies have used these above-mentioned characteristics to develop sea breeze identification methods for individual coastal sites, based on both observations and atmospheric model data. These methods can be applied to analyze local sea breeze processes, or to construct regional climatologies of their occurrences. Identification methods are usually based on local features of the circulation such as the timing, intensity, and typical prevailing conditions (for example, Masselink and Pattiaratchi, 2001; Azorin-Molina et al., 2011a; Soderholm et al., 2016; Masouleh et al., 2019; Arrillaga et al., 2020; Xia et al., 2022), with these sometimes referred to as "filtering" methods (Azorin-Molina et al., 2011b). These are often optimized for small regions, usually with consistent coastline orientations that can be used to subjectively define the onshore and offshore wind sectors. This limits their applicability to continental-scale or global studies of the sea breeze, although Huang et al. (2025b) were able to apply such methods to the entire coastline of China. Despite this, there is a need for more robust sea breeze identification methods that are based on general physical characteristics.

Several studies have defined sea breezes using robust, physically-based methods, although their applicability to general identification have not been fully explored. For example, Coceal et al. (2018) present an objective method for detecting sea breeze fronts from weather station data, based on the local time rate of change in temperature and moisture, as well as wind speed, direction and gust. Hallgren et al. (2023) propose a method for sea breeze forecasting based on the identification of coastline-relative circulations in the planetary boundary layer, using data from numerical weather prediction models. However, these methods have only been tested for a selection of coastal sites in the United Kingdom and Sweden, respectively. Other studies have utilized diagnostics of the dynamical flow, such as convergence in the wind field or frontogenesis methods, for the identification of mesoscale fronts from numerical model data and observations, including sea breezes (Kraus et al., 1990; Arnup and Reeder, 2007; Jenkner et al., 2010; Grant and Van Den Heever, 2014). However, these types of methods have not yet been applied to general sea breeze identification over a large region. Previous studies have also noted that robust sea breeze identification is complicated by other mesoscale fronts and circulations that can exist in coastal regions, such as convective cold pools or topographically-forced flows (Cafaro et al., 2019; Hallgren et al., 2023). It may be difficult or potentially impossible to separate sea breezes from these other types of circulations, given their potential interactions.

Here, we develop software for application of three different physically-based diagnostics to identify sea breezes from atmospheric model data. These diagnostics are tested over the whole of Australia for a 6-month period. Each of these diagnostics has been developed or applied by previous studies on sea breezes. This includes, as discussed above, the "sea breeze index" developed by Hallgren et al. (2023), the time rate of change method developed by Coceal et al. (2018), and the frontogenesis parameter (for example, Thomas and Schultz, 2019). The use of multiple diagnostics here allows for a quantification of uncertainty when assessing results. In addition, we develop our own general "filtering" methods that will attempt to remove non-sea-breeze fronts and circulations. These diagnostics and filtering methods will be described later in this paper. The goal of this study is to demonstrate these new methods and software, as well as to investigate the potential of the diagnostics for robust sea breeze detection over Australia, while documenting potential sources of uncertainties and mis-classifications.

In addition, these sea breeze diagnostics will be applied to four different atmospheric model datasets, to investigate the impact of model resolution on sea breeze representation and identification. We will use a common 6-month period of data during the Austral summer, from the ERA5 reanalysis (around 25 km horizontal grid spacing), the Bureau of Meteorology Atmospheric Regional Reanalysis for Australia (12 and 4 km grid spacing), and a regional configuration of the Australian Community Climate and Earth System Simulator (AUS2200, 2.2 km grid spacing). It is assumed that sea breeze representation will vary between these model datasets, as previous studies have shown that km-scale models provide added value of sea breeze representation over coarser-scale models (Cafaro et al., 2019). The accuracy of sea breeze detection methods and model dataset representations will be qualitatively assessed through the analysis of individual sea breeze cases, as well as through statistics of sea breeze occurrences over the 6-month period, compared with theoretical understanding, previous studies, and observational datasets.

This paper is structured as follows: Firstly, the model datasets and observations are introduced (Section 2). Then, the methods for diagnosing sea breezes and defining sea breeze objects are described (Section 3). We then present six individual cases for three different regions of Australia from January 2016 (Section 4), including cases where the identification performs as expected, as well as cases where there are potential mis-classifications. Next, statistics of sea breeze occurrences are presented over Australia for the 6-month study period, including the spatial and diurnal distribution of objects (Section 5). Finally, a discussion and conclusion (Section 6) are presented.

#### 2 Datasets

#### 75 **2.1 ERA5**

The ERA5 reanalysis is produced by the European Centre for Medium Range Weather Forecasts (Hersbach et al., 2020), with data here obtained from the Australian National Computational Infrastructure (NCI) and the Analysis-Ready, Cloud Optimized archive (ARCO, Carver and Merose (2023)), both provided on a  $0.25^{\circ}$  latitude-longitude grid. Variables include the u and v wind components and geopotential height for the first 38 model levels up to a height of approximately 4,300 m above the surface, 10 m u and v winds, 2 m temperature, and boundary layer height. In addition, 2 m dew point temperature and surface pressure are used to compute specific humidity, using the metpy Python package (May et al., 2019). Each of these variables are available at hourly instantaneous intervals. Before computing the relevant sea breeze diagnostics presented in Section 3.2, the model-level wind data is linearly interpolated to regular height levels above the surface at 100 m spacing using the geopotential height field, up to a height of 4500 m (with some extrapolation between 4300 and 4500 m), which is expected to include the planetary boundary layer.

# 2.2 BARRA

The Bureau of Meteorology Atmospheric Regional Reanalysis for Australia version 2 (BARRA2, Su et al. (2022)) has an atmospheric model component (Australian Community Climate and Earth-Systems Simulator, ACCESS) based the U.K. Met

https://doi.org/10.5194/egusphere-2025-4848 Preprint. Discussion started: 14 November 2025

© Author(s) 2025. CC BY 4.0 License.

Office Unified Model. The model is configured with approximately 12 km grid spacing over the wider Australasian region, in a regional configuration called BARRA-R2, and assimilates a range of relevant regional observations, with ERA5 providing lateral boundary conditions. BARRA-C2 (Su et al., 2024) is downscaled from BARRA-R2, and is configured with around 4 km grid spacing. BARRA-C2 is designed to be convection-permitting, and so it does not use a convection parameterization, in contrast to BARRA-R2.

BARRA2 data is hosted on the Australian NCI and includes the 10 m *u* and *v* wind components, 1.5 m specific humidity and 1.5 m air temperature, representing instantaneous hourly values. Model level output was not available for use in this study, and so one of the sea breeze diagnostics is not applied to the BARRA datasets (the sea breeze index, see Section 3.2.1).

For BARRA-C2, it was decided that spatial smoothing was required to reduce the impact of small-scale gradients on sea breeze identification methods, similar to the study of Jenkner et al. (2010) for identification of mesoscale fronts in the European Alps. A Gaussian smoothing filter is applied in the latitude and longitude dimensions (Virtanen et al., 2020), with a standard deviation of 2 judged to sufficiently remove small spatial gradients while preserving the structure of sea breeze fronts, based on manual inspection of individual cases.

## 2.3 AUS2200

100

105

110

AUS2200 is a separate configuration of the ACCESS model with 2.2 km grid spacing (Huang et al., 2025a). AUS2200 uses ERA5 lateral boundary forcing, but without data assimilation. Three, two-month AUS2200 simulations are used, over January–February 2013, 2016, and 2018. These runs were produced in support of a separate study on rainfall in the Australian tropics, and were chosen for that study based on three active Madden-Julian Oscillation events during different phases of the El Niño-Southern Oscillation (Dao et al., 2025). The simulations cover the whole of Australia, with the temporal extent used here to define a study period of 6 months. Each simulation takes place during the Austral summer period, when frequent sea breeze circulations are expected.

For computing sea breeze diagnostics, model level u and v winds are used up to a height of 4500 m (36 levels), as well as 10 m u and v winds, boundary layer height, and specific humidity and air temperature on the lowest model level (5 m). Model-level data are provided at unevenly-spaced hybrid-height levels, and are interpolated onto the same evenly-spaced vertical grid as for ERA5 (100 m intervals to a height of 4500 m). Hourly instantaneous values are used for wind, temperature and humidity data. However, for boundary layer height, only hourly mean values are available.

Similar to BARRA-C (Section 2.2), smoothing is applied to the AUS2200 fields prior to calculating sea breeze diagnostics. A Gaussian filter is applied in the latitude, longitude and vertical dimensions, with a standard deviation of 4. The sensitivity of sea breeze identification to this smoothing is explored in the Supplementary Material (Section S3), where it is found that the choice of smoothing can impact the number of objects identified, although the diurnal and spatial structure of object statistics is relatively unaffected.

140

#### 120 2.4 Observations

Two observational datasets are used to evaluate the model representation of sea breezes and the sea breeze identification methods. Firstly, visible Himawari imagery (channel 3 corrected reflectance, Bureau of Meteorology (2021)) is analyzed for individual cases in Section 4. Sea breezes can often be seen in visible satellite imagery by linear cumulus cloud features aligned with the coast (Planchon et al., 2006). These features are used here to evaluate if a sea breeze was present in reality. Secondly, hourly automatic weather station (AWS) data from the Australian Bureau of Meteorology are used to evaluate the model representation of sea breeze fronts. This is done by calculating one of the sea breeze diagnostics from the station data (see hourly rate of change method, Section 3.2.3), and comparing the spatial correlation of potential sea breeze occurrences between the observations and the closest model points, for a set of 712 stations over Australia.

#### 3 Sea breeze identification methods

The sea breeze identification method is shown schematically in Figure 1. The software that applies these methods is split into three modules: Firstly, the pre-processing of the atmospheric model datasets, including the projection of wind components to coast-relative coordinates (see Section 3.1), and the smoothing of model fields. Secondly, three sea breeze diagnostics are calculated, as described in Section 3.2. Finally, the diagnostics are converted to sea breeze objects, in a process described in Section 3.3.

#### 135 3.1 Defining coastline angle of orientation and the onshore wind component

A method to calculate coastline orientations throughout the study domain is required to define onshore and offshore wind speeds. We present a method applied to all grid points using each model's land-sea mask, where the direction to all nearby coastline points is calculated to find an average coastline direction. The average coastline direction is weighted, such that each grid point is most strongly influenced by the nearest coastline points, with a decreasing influence with distance from the grid point. Then, the average coastline direction is rotated clockwise by  $90^{\circ}$  at each grid point to define an average coastline orientation,  $\theta$ . The method for calculating  $\theta$  is described in detail in the Supplementary Material (Section S1), including the weighting functions.

Once  $\theta$  has been estimated, then the onshore wind component, u', can be calculated for each grid point from the u and v wind components, as:

$$u' = v \times \cos(90^{\circ} - \theta) - u \times \sin(90^{\circ} - \theta) \tag{1}$$

where positive u' is defined as an onshore wind, and negative u' is defined as an offshore wind.

The coastline orientations ( $\theta$ ) are shown in Figure 2 for AUS2200 and ERA5. Additional detail can be seen based on the AUS2200 land-sea mask compared with ERA5, particularly for small islands that are not resolved in ERA5 (for example, between the mainland of Australia and the island of Tasmania at around 40°S). In addition, the weighted variance of the set of coastline angles at each point is calculated as an estimation of uncertainty in  $\theta$ . Large variance indicates multiple coastlines of

**Figure 1.** Flowchart demonstrating the sea breeze object identification method. Blue boxes represent inputs from model datasets, orange boxes represent derived quantities or functions.

varying orientation influencing a grid point. The weighted variance of coastline angles is also shown in Figure 2. A value of 0.5 appears to highlight regions where dominant coastline orientations are not able to be defined, such as halfway in between two land masses, or in the middle of an island. This value will be used in further sections to indicate regions with high coastline orientation uncertainty.

## 155 3.2 Sea breeze diagnostics

Three different diagnostics are used to identify sea breezes from model data. These are the sea breeze index (SBI), the moisture frontogenesis parameter (F), and an algorithm that combines the hourly rate of change in temperature, moisture, and onshore wind speed (referred to here as H). These diagnostics are intended to characterize the sea breeze based on either a front (F) and (F) or circulation (SBI), using complementary methods. For example, the SBI uses three-dimensional wind data, (F) uses

Figure 2. The weighted average angle between each point and the coastline, representing average orientation orientation ( $\theta$ ), shown separately for (a) ERA5 and (b) AUS2200. Also shown is the coastline orientation variance at each grid point, for (c) ERA5 and (d) AUS2200. Variance values above 0.5 are highlighted here using stippling, as regions where it may not be appropriate to define a coastline orientation angle. In each panel, the coastline is shown as a contour between land and ocean in the land sea mask. In panel (d), the location of the case study domains (Section 4) are indicated with black dashed boxes.

https://doi.org/10.5194/egusphere-2025-4848 Preprint. Discussion started: 14 November 2025

© Author(s) 2025. CC BY 4.0 License.


horizontal gradients in wind and moisture, and *H* uses a time series approach. Because *H* is a time series method, it can also be applied to station observations and used as a model evaluation tool. This is explored in the Supplementary Material (Section S2), where it is shown that the models can represent spatial variations in observed hourly local changes, as represented by the *H* diagnostic, with a high degree of accuracy, although this depends on model resolution to some extent.

Each diagnostic will be applied to the four different atmospheric model datasets over Australia described previously, except where data is unavailable (there are some limitations for the *SBI* where model-level data is required, and unavailable for BARRA). Each diagnostic has been developed or applied by previous sea breeze studies, and the novel aspect here is their application to all model grid points in a continental-scale domain with varying coastlines, for several different model datasets.

## 3.2.1 Sea breeze index, SBI

The Sea Breeze Index (*SBI*) is defined here following Hallgren et al. (2023). The *SBI* is based on the identification of a circulation with onshore flow near the surface and offshore flow at a higher level within the boundary layer, relative to a coastline with an angle of orientation  $\theta$  (angle from north). The method requires a vertical resolution that can sufficiently resolve mesoscale circulations in the boundary layer, as well as model output on all vertical levels rather than standard pressure levels, which may be too coarse to capture the offshore flow aloft. The *SBI* is calculated at each grid point where the following conditions are met, where  $\alpha$  is the low-level wind direction and  $\beta$  is the upper-level wind direction (otherwise the *SBI* takes a value of 0):

- 1.  $\alpha$  is between  $\theta$  and  $\theta + 180^{\circ}$  (near-surface flow is onshore)
- 2.  $\beta$  is between  $\theta 180^{\circ}$  and  $\theta$  (upper-level flow is offshore)
- 3.  $\beta$  is within  $\pm 90^{\circ}$  of  $\alpha + 180^{\circ}$  (the near-surface and upper-level flow are opposing)

If these three conditions are true, the SBI is calculated as:

$$SBI = cos[\alpha - (\theta + 90^{\circ})] \times cos[\alpha + 180^{\circ} - \beta]$$
 (2)

For further details of this method, the reader is referred to Hallgren et al. (2023). The SBI ranges from 0 to 1 and is maximized when the onshore flow is perpendicular to the coastline and the upper-level flow is in the opposing direction. Following the recommendations of Hallgren et al. (2023),  $\alpha$  is defined at a height of 10 m, and  $\beta$  is defined separately for each model level within the planetary boundary layer. The maximum SBI in the vertical is then retained for each grid point. For application to all grid points in the Australian domain,  $\theta$  needs to be defined by an objective method, which is described in Section 3.1 and in the Supplementary Material (Section S1).

As discussed in Hallgren et al. (2023), there are several limitations to the SBI. These include the potential for false identification of sea breezes due to other sources of wind direction changes in the boundary layer, such as convective cold pools, synoptic-scale fronts, and low-level jets. SBI values may also be impacted by background synoptic-scale surface winds and vertical wind shear, given that the total wind direction is used to define a circulation rather than perturbations to the flow.

# 3.2.2 Moisture frontogenesis, F

Frontogenesis (F) can be defined as the time rate of change in the gradient of a scalar quantity, representative of frontal growth. Pettersen (1956) showed that assuming a conserved scalar, S, and considering changes in its gradient at the surface due only to horizontal kinematic processes, F can be expressed in terms of horizontal flow deformation and divergence:

$$F = \frac{1}{2} |\nabla_h S| [D_{total} \times cos(2B) - \delta]$$
 (3)

where  $D_{total}$  is the total deformation, B is the angle between the axis of dilatation and the isopleths of S, and  $\delta$  is the divergence.  $D_{total} = \sqrt{D_{stretch}^2 + D_{shear}^2}$ , where  $D_{stretch} = du/dx - dv/dy$  is the stretching deformation and  $D_{shear} = dv/dx + du/dy$  is the shearing deformation. Following Arnup and Reeder (2007), who examine dry lines in northern Australia including the role of sea breezes, specific humidity (q) is chosen as the scalar quantity, with F therefore representing moisture frontogenesis. F is calculated using MetPy (May et al., 2019) with an implementation based on Bluestein (1993), using u and v winds at a height of 10 m and near-surface q.

There are potential limitations in the application of F to sea breeze identification. Besides sea breezes, F may also identify other moisture fronts, such as dry lines, synoptic-scale fronts, and convective cold pools. In addition, the magnitude of horizontal gradients in q will be very sensitive to the resolution of the model grid. Therefore, for defining sea breeze objects from F, model-dependent thresholds will be used (see Section 3.3).

#### 3.2.3 Hourly rate of change, H



The local time rate of change in moisture, air temperature, and onshore wind speed is used to diagnose the passage of the sea breeze front, following the method of Coceal et al. (2018). The method combines each time rate of change into a single diagnostic, based on a fuzzy-logic approach. The diagnostic will be applied to hourly model output and referred to as the hourly rate of change diagnostic (*H*).

Firstly, the hourly rate of change time series' in near-surface specific humidity (q) and air temperature (t), and 10 m onshore wind speed (u'), are calculated for each point on the model grid using a backward difference. The procedure for calculating u' from the u and v wind is described in Section 3.1. For t, the negative of the time series is taken so that the passage of the sea breeze front (a drop in temperature) is defined as an increase in sea breeze front potential, consistent with q and u'. The hourly rate of change for each quantity is then converted to a fuzzy-logic function, f(x):

$$f(x) = \begin{cases} y_1 & x \le x_1 \\ y_1 + (\frac{y_2 - y_1}{x_2 - x_1})(x - x_1) & x_1 






a value that is a linear interpolation between  $y_1$  and  $y_2$ . The values of  $y_1$ ,  $y_2$ ,  $x_1$  and  $x_2$  are chosen following Coceal et al. (2018), with  $y_1 = 0$ ,  $y_1 = 1$ ,  $x_1 = 0$ , and  $x_2$  as half of the maximum value of x over all locations. The f(x) for each quantity is calculated in monthly time chunks, so that the value of  $x_2$  may change slightly for different periods. However, this is not expected to impact the results, as f(x) is not sensitive to the exact values of  $x_1$  and  $x_2$  (Coceal et al., 2018). Then, f(q), f(t) and f(u') are averaged together to give a time series of H at each grid point.

There are some minor differences between H as defined here and in Coceal et al. (2018). This includes the use of an hourly rate of change instead of 10-minute intervals, while u' is used here rather than wind speed, direction and gust strength, for simplicity. There are also uncertainties in the application of this method to broader regions, given that it was developed for the London, U.K. region. Similar to F, there is potential to incorrectly classify other atmospheric fronts as sea breezes. This will be addressed to some extent through additional filtering, described in the next section.

# 3.3 Sea breeze objects and filtering

After the diagnostics *F*, *H*, and *SBI* are calculated, thresholds are used to convert them to binary fields, which are then used to define candidate sea breeze objects. A candidate sea breeze object is defined as any connected group of pixels where a diagnostic exceeds a certain threshold, with diagonal pixels considered connected. The thresholds are based on percentiles, allowing them to change with each dataset according to the distribution of the diagnostic. This is relevant for *F*, given that spatial gradients are sensitive to the grid spacing of the model data. Here, the 99.5<sup>th</sup> percentile of each diagnostic was used as a threshold, over all grid points throughout the Australia region for the 6-month study period, as reported in Table 1. This percentile was chosen based on examining representative sea breeze cases, where it was judged to provide an appropriate balance between detection of sea breezes and ignoring a large proportion of false alarms. However, this threshold could be changed for different applications. The sensitivity of results to the choice of percentile threshold is explored in the Supplementary Material (Section S3) for the *F* diagnostic, suggesting that lower thresholds may indicate too many coastal objects that are not sea breezes.

The candidate sea breeze objects are then filtered based on several criteria related to their morphology and atmospheric conditions. These filters represent somewhat arbitrary choices for our application over the entire Australian continent, and may not represent optimal choices for other more specific applications. In the *sea\_breeze* (v1.1) code provided for identification (Brown et al., 2025), settings are available for flexible thresholds and filters. The sensitivity of object detection to each of the filters is analyzed systematically in the Supplementary Material (Section S3), and discussed qualitatively in relation to individual case studies (Section 4). Objects are filtered based on their associated area, aspect ratio, orientation, onshore wind speed and land-sea temperature contrast. Specifically, sea breeze objects are only retained if they are at least 12 pixels in size, corresponding to a minimum area ranging from around 40 km² in AUS2200 to around 6,500 km² in ERA5. This area criteria is applied on a pixel basis to account for larger circulations being resolved in coarser-scale models, with a value of 12 thought to represent a reasonable lower-bound for the size of a sea breeze object in the highest-resolution model, AUS2200, and larger than the expected effective resolution, usually proposed to be approximately 7 pixels (Skamarock, 2004).

An aspect ratio threshold of 2:1 is imposed, as it is assumed that the sea breeze should generally be oriented along the coastline, and therefore much longer in the along-shore direction compared with the cross-shore direction. This is intended to



Table 1. The 99.5th percentile values of each diagnostic for each model, over the entire study domain and period

| Model   | SBI  | F (g/kg/100 km/3h) | Н    |
|---------|------|--------------------|------|
| ERA5    | 0.31 | 4.32               | 0.19 |
| BARRA-R | -    | 12.68              | 0.17 |
| BARRA-C | -    | 16.10              | 0.21 |
| AUS2200 | 0.41 | 18.80              | 0.20 |

filter out objects that are relatively circular, such as cold pools from cellular convection that spread out radially at the surface. However, the aspect ratio filter may also potentially remove sea breezes along coastlines with small horizontal scales, such as small islands, as discussed later for individual cases (see Section 4.2). Objects are also filtered out if their angle of orientation differs significantly from the orientation of the associated coastline (defined by  $\theta$  in Section 3.1), to try and remove objects that are not primarily forced by coastal processes. A relatively weak tolerance of 45° is used, to allow for the deformation of sea breeze objects as they move onshore, although it is acknowledged that this tolerance could potentially be exceeded in cases with strong horizontal wind shear, as well as in situations where an object propagates to an adjacent coastline with a significantly different orientation (from one side of a peninsula to another, for example). Both the aspect ratio and orientation of the sea breeze objects are estimated using an ellipse fitted to the object, using the scikit-image python package (Van Der Walt et al., 2014).

In addition, candidate objects are removed when the average wind speed is not onshore or when the land-sea temperature contrast is non-positive. The average onshore wind speed is estimated by first calculating the onshore component of the near-surface wind, u' (see Section 3.1), and then averaging it over the area of the object. Objects are then removed when the onshore wind speed is non-positive, with the expectation that the sea breeze should have an onshore surface wind component. The land-sea temperature contrast,  $\Delta T$ , is calculated for each grid point as the difference in the near-surface air temperature between the closest ocean point and the maximum temperature within 50 km of the closest land point. A 50 km maximum is applied over the land, considering land points only, so that  $\Delta T$  is not impacted by localized cooling due to the sea breeze front. This distance was chosen to capture grid points sufficiently ahead of the front that are still close enough to influence the land-sea temperature forcing. For each object, the land-sea temperature contrast is then defined by the average  $\Delta T$  over the object.

Filtering settings are summarized in Table 2. We have chosen not to constrain the objects based on the time of day, distance from the coast, or prevailing wind conditions, with the aim of allowing the method to be as robust as possible.

#### 4 Example cases

A selection of cases are presented from January 2016, to demonstrate the identification of sea breezes using the methods and models outlined above. Cases are shown for three regional domains around Australia, including a mid-latitude domain (Perth,


Table 2. Filtering conditions for sea breeze objects. Candidate sea breeze objects are retained if they meet all criteria.

| Filter                                | Conditions                                |  |
|---------------------------------------|-------------------------------------------|--|
| Orientation relative to the coastline | Within $45^{\circ}$                       |  |
| Aspect ratio                          | Greater than 2                            |  |
| Area                                  | At least 12 pixels                        |  |
| Land-sea temperature difference       | Must be positive (land warmer than ocean) |  |
| Onshore wind speed                    | Must be positive                          |  |

Section 4.1), a tropical domain with complex coastlines (Darwin, Section 4.2), and a midlatitude domain with significant topography near the coast (Southeast Australia, Section 4.3). The location of each domain is shown in Figure 2d.

For each domain, two types of cases are shown: a sea breeze identified consistently between multiple methods and models, providing confidence in the methods, and a case where there are inconsistencies. These latter cases are intended to demonstrate where there are uncertainties in the methods, while leading to a discussion around the ultimate objectives of how sea breezes should be characterized and separated from other fronts and circulations.

## 4.1 Perth (6 and 11 January, 2016)

Figure 3 demonstrates a case for the Perth domain on 6 January 2016 (16:00 local time). In this case, most models and methods indicate a clear sea breeze, which at this time has already moved onshore. The only exception is the H diagnostic applied to ERA5, where values do not exceed the relevant threshold for object identification. Each model represents the associated onshore flow behind the sea breeze front, with calm winds or offshore flow ahead. The presence of a sea breeze is supported by the station wind vectors and Himawari visible satellite imagery. This includes onshore flow at the coast and calm winds inland, with a strip of cloudless air along the coast and a line of cumulus cloud along the sea breeze front. This provides confidence in the application of these diagnostics to well-defined sea breeze cases. There are some inland gradients in wind and moisture that provide elevated values of F and H ahead of the sea breeze front in AUS2200, likely due to convection as suggested by the visible satellite image, although these are mostly filtered out due to the morphology of the objects.

Figure 4 shows a case on 11 January 2016 (04:00 local time), where a synoptic-scale cold front is incorrectly identified by the *F* and *H* diagnostics as a sea breeze object by some of the models. This occurs in the early morning, which is too early for the sea breeze to form. The object is not filtered out in all cases, given that it is oriented in a large line parallel to the coastline. The potential to misclassify synoptic-scale fronts should be kept in mind as a limitation of methods that characterize the sea breeze as a front. These frontal objects could potentially be removed by additional filtering, based on a time-of-day requirement, or a local increase in specific humidity (synoptic-scale cold fronts are dry, in contrast to the sea breeze front).

**Figure 3.** Sea breeze diagnostics for 2016-01-06 08:00 UTC (16:00 local time) for a Perth sea breeze case, including (a–d) F, (e–h) H, and (i and j) SBI, calculated from (a, e, and i) AUS2200, (b and f) BARRA-C, (c and g) BARRA-R, and (d, h, and j) ERA5. Model 10 m wind vectors are shown in each panel, and compared with (l) station wind observations. Sea breeze objects are shown with a black contour. The corresponding Himawari visible satellite image is shown in (k), with the BARRA-C F sea breeze mask indicated with an orange contour for comparison. In each panel, the model land-sea mask is also contoured with a black line, representing the coastline. Stippling indicates regions where there is no dominant onshore direction (see Section 3.1).

Figure 4. As in Figure 3 but for 2016-01-11 20:00 UTC (04:00 local time). There is no visible satellite imagery available for this case.






# 4.2 Darwin (10 and 12 January, 2016)

300 Figure 5 demonstrates a case on 10 January 2016 (14:30 local time) for the Darwin domain. Each of the models in Figure 5 indicate local onshore flows around most of the coastline, likely representative of sea breezes. These onshore winds are also observed in the station observations. There is some evidence for sea breeze fronts in the visible satellite image, with cloud features oriented along the coastline. However, there are also regions of active deep convection along the coast, such that the identification of sea breezes from the imagery is unclear.

Many of these onshore flows are indicated as sea breeze objects using the F diagnostic, except when applied to the ERA5 model. The SBI also produces some sea breeze objects around the coast from AUS2200, with this consistency between F and SBI providing confidence that sea breezes are being represented by the model, and that each of these diagnostics is appropriate for identification. In contrast, the H diagnostic appears limited in this case, but does produce sea breeze objects several hours later (not shown). This may be related to the tendency of H to identify sea breeze fronts later in their life cycle (explored later in Section 5).

There are high values of F over the Tiwi Islands to the north of Darwin (11.5°S, 131°E) in AUS2200, BARRA-C and BARRA-R, with converging onshore flows that have been well-documented in previous studies on sea breezes in this region (for example, Oliphant et al., 2001). However, only BARRA-C has a sea breeze object over the islands, whereas these values are filtered out in AUS2200 and BARRA-R. This highlights potential limitations of applying the same set of filters to the entire continental-scale region, where small-scale circulations such as on small tropical islands may be filtered out.

For the case shown in Figure 6 on January 12 (15:30 local time), there are local onshore flows along much of the coastline in the models and station observations. Some of these regions are highlighted as sea breeze objects using F applied to AUS2200 and BARRA-C, as well as the SBI from AUS2200. BARRA-R and ERA5 do not produce strong enough fronts for sea breeze identification with the F or H diagnostics, although there are some objects detected by the SBI applied to ERA5.

Sea breeze identification is complicated in this case by the presence of deep convection throughout the domain, as represented by the convection-permitting models (AUS2200 and BARRA-C), and visible in the satellite imagery. This convection produces cold pools and low level circulations over inland regions that are incorrectly identified as sea breezes by each of the diagnostics in the convection-permitting models. Convective cold pools are potentially very difficult to separate from sea breezes, particularly when they are oriented in a line parallel to the coastline. This is complicated by the fact that the sea breeze front can initiate convection, with transitions from sea breezes to convective cold pools that may appear very similar. It could be argued that the cold pools associated with the sea-breeze-initiated convection are part of the sea breeze itself. There is no explicit separation of these processes in our methods, although once the sea breeze becomes sufficiently modified by convection, it is unlikely to satisfy all of the filtering criteria.

Figure 5. As in Figure 3 but for 2016-01-10 05:00 UTC (14:30 local time) for a case over the Darwin region.

Figure 6. As in Figure 3 but for 2016-01-12 06:00 UTC (15:30 local time) for a case over the Darwin region.




# 4.3 Southeast Australia (9 and 14 January, 2016)

Figure 7 shows a case along the Southeast Australia domain on 9 January 2016 (14:00 local time). This is a region where the topography of the Great Dividing Range is close to the coastline, and hatching on Figure 7 is used to indicate where the topography exceeds a height of 500 m. There are sea breezes identified by each model using the *F* and *SBI* diagnostics, with associated onshore flow, although there are no objects indicated by *H* at this time. There are some objects identified by *H* several hours later (not shown), and analysis in Section 5 will show that this diagnostic reaches larger values later in the sea breeze life cycle, in general.

It is unclear in this case whether the circulations and fronts identified by these methods are forced by land-sea temperature contrasts, topographic heating, or both. Topographic heating and associated up-slope flow occur during the afternoon, and in this region will act in the same direction as the sea breeze, so these may be inseparable. The objects in this case are first detected some distance inland, suggesting that the topography is playing at least some role. This can also be seen in the spatial distribution of sea breeze occurrence frequency over the 6-month study period for this region, shown in Section 5, where there is a local maximum in object counts shifted inland from the coast. This has been observed previously by Clarke (1983), who documented that the sea breeze front can only be detected at a distance of around 60 km inland in this region, potentially due to topographic influences. There are some cumulus cloud features in the satellite imagery associated with the sea breeze objects for this case, although again, it is not clear whether these are topographically forced or initiated by the sea breeze.

The case shown in Figure 8 on 14th January (09:00 local time) is highlighted as a potential mis-classification of sea breezes based on the *F* diagnostic when there is moist onshore flow. There is a sea breeze object highlighted by AUS2200 and BARRAR using the *F* diagnostic along the coast that is not present in any of the other diagnostics, or evident in the satellite observations. This case occurs during the early morning when sea breezes are not expected to occur. It should be therefore kept in mind that large values of *F* can be produced when there is strong onshore flow and convergence, with no sea breeze front present. Future work may be able to eliminate these mis-classifications through additional filtering, including a local time criteria or tracking objects to identify stationary fronts near the coastline.

**Figure 7.** As in Figure 3 but for 2016-01-09 04:00 UTC (approximately 14:00 local time) for a sea breeze event over the south-east of Australia. Shaded hatched regions indicate model topography that exceeds 500 m.

**Figure 8.** As in Figure 3 but for 2016-01-14 23:00 UTC (approximately 09:00 local time) for a sea breeze event over the south-east of Australia. Shaded hatched regions indicate model topography that exceeds 500 m.








# 5 Statistics of sea breeze object occurrences

## 5.1 Spatial distribution

We present statistics of sea breeze occurrence frequency, defined as the fraction of days in the 6-month January–February study period with a sea breeze object at each grid point (from 00:00 to 23:00 UTC). The spatial distribution of sea breeze occurrence frequency is presented in this section for each available model dataset, and each diagnostic.

Each of Figures 9–11 indicate that sea breeze objects are almost exclusively identified over land. The spatial distribution of sea breeze objects shows a clear maximum along the coastline for the F (Figure 9) and SBI (Figure 10) diagnostics. Although this is a basic expectation of sea breezes, there was no constraint placed on the distance from the coastline in any of the diagnostic or filtering methods.

The *F* diagnostic produces a higher occurrence frequency of sea breeze days in most regions along the coastline compared with the *SBI*, while the SBI appears to produce a higher proportion of objects over far inland regions (likely mis-classifications). Both *F* and the *SBI* indicate local maxima in occurrence frequency on the northwest coast of the Australian continent for this 6-month period, as well as along the Gulf of Carpentaria coast (around 18°S, 140°E), Eyre Peninsula (around 33°S, 135°E) and southeastern Australia (around 35°S, 150°E). This consistency in spatial distribution between circulation and frontal diagnostics provides confidence in the accuracy of the identification methods. For the *SBI* and *F* diagnostics, the local maxima in southeastern Australia is displaced inland from the coast towards the relatively high topography of this region, indicating that the topography may be playing a role in the forcing and/or identification of sea breeze objects, with these processes being difficult to separate. For both diagnostics, the higher-resolution models (AUS2200 and BARRA-C) tend to produce more frequent occurrences of sea breeze objects, with a distribution that extends further inland, compared with the coarser models (BARRA-R and ERA5).

There is a significantly higher number of sea breeze objects identified by *F* along the west coast of Australia compared to the east coast. This is also present to a lesser extent in results for the SBI. This east-west asymmetry in sea breeze occurrences could be expected due to warmer near-surface temperatures over the land in the western part of the continent during January–February, and a stronger associated land-sea temperature contrast for sea breeze forcing. This is demonstrated in Figure 12, which shows the average daily maximum near-surface air temperature over the study period. There is a significantly higher land-sea temperature gradient based on the daily maximum temperature along the west and north west coast, compared with the east coast, shown in each of the model datasets. This is mainly driven by temperatures over the land, but there are also some differences over the ocean, associated with the presence of the East Australia Current (off the east coast of Australia).

Our application of the H diagnostic to all grid points in the domain results in a spatial distribution of objects that is shifted inland from the coastline across Australia (Figure 11). The shift may be due to larger local changes in temperature and humidity later in the day when the sea breeze front has moved inland, into regions where the land surface is relatively dry and warm due to daytime solar heating. This will be investigated further in the next section. The application of this diagnostic may need to be restricted to coastal locations for more targeted identification of sea breezes earlier in their life cycle. Other than ERA5, there are local maxima in the spatial distribution for H that are similar to F and the SBI. However, there are additional regions of

Figure 9. The fraction of days with a sea breeze object from 00:00 to 23:00 UTC at each pixel, based on the F diagnostic over the entire 6-month study period. Shown separately for each model dataset. Regions exceeding 500 m above sea level are indicated with a contour, and stippling indicates regions without a dominant coastline direction (see Section 3.1). Note that the color scale is not evenly spaced, to capture the spatial variability and small occurrence frequencies.

**Figure 10.** As in Figure 9 but for SBI

high occurrence frequencies, such as along the southern coast of the Australian mainland, that are more prominent compared with the other diagnostics.

**Figure 11.** As in Figure 9 but for H

**Figure 12.** Average daily maximum near surface temperature from each model dataset over the study period (January–February 2013, 2016 and 2018).

Figure 13. The mean occurrence frequency of sea breeze objects at each hour of the day in local time (vertical axis) and at different distances from the coast (horizontal axis) for each diagnostic and model dataset. Note that there are no SBI data available for the BARRA datasets. Occurrence frequencies are averaged over bins spaced at 25 km intervals from the coast, with positive values representing onshore locations. The mean onshore wind (u') perturbation from the daily mean is shown with a dashed contour, for comparison, with contours of 0.25, 0.50, 0.75, and 1.0 m/s.

## Diurnal variability and inland propagation


We now consider the diurnal cycle of sea breeze occurrence frequency, as well as the inland propagation of sea breeze objects. Figure 13 shows the mean sea breeze occurrence frequency for each model, method, and hour of the day, binned into different distances from the coastline. We also compare the sea breeze occurrence frequency in Figure 13 with the mean onshore wind (u', see Section 3.1) perturbation from the daily mean, averaged over the same inland distance bins as for the sea breeze objects. The u' perturbation is intended to represent the average cross-shore surface wind associated with the diurnal cycle. This should primarily include the sea breeze circulation, but may also include other effects such as the diurnal cycle of vertical mixing or topographic flows. 395




According to the results presented in Figure 13, the F diagnostic identifies the sea breeze front from around 10:00 local time, near the coastline, in AUS2200 and BARRA-C. This onset time is slightly later in BARRA-R and ERA5, at around 11:00 local time. The u' perturbation forms on the coastline a couple of hours later, likely representing the surface winds that develop due to the sea breeze circulation. The occurrence frequency of objects based on the F diagnostic reaches a maximum on the coastline at around 13:00 local time in AUS2200 and BARRA-C and 14:00 local time in BARRA-R and ERA5, before moving inland. In AUS2200, the objects as identified by F continue to move inland and appear to reach as far as 400 km from the coast at 23:00 local time. There are also some objects identified by F from 300–450 km inland at around 18:00 local time. It is unclear whether these objects are sea breezes that are propagating faster than the dominant signal, or if these are false detections. A similar distribution of inland objects appears in the BARRA-C and ERA5 models, while BARRA-R tends to have a reduced inland extent. The diurnal timing and inland movement of sea breeze objects using F matches very well with the structure of the u' perturbation, with the u' perturbation lagging behind the sea breeze front. This provides confidence that the distribution of F objects represents the sea breeze signal, assuming the u' perturbation is largely related to sea breezes.

The *H* diagnostic does not appear to identify objects until much later in the sea breeze life cycle, at around 13:00 local time in AUS2200 and BARRA-C and 15:00 local time in BARRA-R and ERA5, with a maximum occurrence frequency around 100 km inland. This could be because of larger local changes in temperature and moisture due to the sea breeze front later in the day at inland locations, which are relatively hot and dry, compared with locations near the coastline. From 13:00 local time in AUS2200, the inland propagation of sea breeze objects based on *H* appears similar to the *F* diagnostic and *u'* perturbation, although objects are detected much further inland, to a distance of around 500 km. This inland distance is even further than the distance inland that the *u'* perturbation travels, defined by the *u'* perturbation contour of 0.25 m/s in Figure 13. There is similar inland propagation in each of the other models in the late afternoon, but with a reduced inland extent compared with AUS2200, and with an unrealistically large amount of objects identified at 19:00 local time that is inconsistent with propagation of the sea breeze signal. The large number of objects that occur at this time are not propagating, based on their occurrence across all locations simultaneously. These objects might therefore be related to other local changes in temperature and wind speed, such as radiative cooling in the early evening, rather than sea breeze fronts. This suggests that the *H* diagnostic may not be applicable for an entire regional domain for these models.

The *SBI* starts to identify sea breeze objects near the coast at 09:00 local time in AUS2200 and 10:00 local time in ERA5, consistent with the *F* diagnostic. There is also a significant number of objects detected at inland locations in the morning in both models, that persist throughout the day. These inland objects are therefore likely not related to sea breeze circulations, but may be representative of other boundary layer circulations, or instances of low-level, directional vertical wind shear. These inland objects can also be seen spatially in Figure 10. The *SBI* does indicate some propagation of sea breeze objects based on the orientation of contours of occurrence frequency originating near the coast, although this propagation does not persist after around 15:00 local time. The *SBI* therefore represents an earlier portion of the sea breeze life cycle compared with frontal objects, consistent with the sea breeze circulation being maximized around the coastline during the period of strongest land-sea temperature contrast.




#### 430 6 Discussion and conclusion

We have tested three sea breeze diagnostics in their application to atmospheric model data over a continental-scale domain, with the aim of investigating the potential for a robust, physically-based identification method. These diagnostics characterize the sea breeze as either a front or circulation, and additional filtering is applied in an attempt to remove other mesoscale and synoptic-scale phenomena. This includes the requirement for objects to be oriented along the coastline with onshore winds, with the coastline orientation and onshore wind component defined using a new objective method (Section 3.1 and Section S1 of the Supplementary Material). The diagnostic and filtering methods do not make any assumptions about the large-scale background wind or the timing and intensity of local sea breezes for individual coastal sites. This is in contrast to other regional identification methods that are usually based on several local characteristics of the sea breeze (see for example the review by Azorin-Molina et al., 2011b). The software for these diagnostic and filtering methods are available in an open source repository (sea\_breeze v1.1, Brown et al. (2025)).

We have demonstrated that the diagnostic and filtering methods can capture individual sea breeze cases (Section 4), with a spatial and diurnal distribution of objects over the 6-month study period in agreement with expectations of the sea breeze (Section 5). This includes a maximum in object occurrences around the coastline, with inland propagation consistent with the movement of the average onshore wind perturbation relative to the daily mean (Figure 13). We also found a deep inland propagation of sea breeze fronts, up to around 400 km according to the AUS2200 model using the frontogenesis diagnostic (*F*). Such deep inland propagation has been observed previously for Australia, particularly for lower latitudes (Clarke, 1983).

We found similarities in the spatial and diurnal distribution of sea breeze object occurrences between the frontogenesis (*F*) and sea breeze index (*SBI*) diagnostics, representing two complementary methods based on the sea breeze front and circulation, respectively. We also found consistency between these methods for individual cases in very different regions of Australia, in agreement with observational evidence from satellite and station wind data. The consistency between these two complementary methods provides confidence in their potential application.

However, despite the identified objects here displaying sea breeze characteristics, is also clear that sea breezes can be difficult to separate from other frontal objects and circulations near the coast in km-scale model output. These may include convective cold pools, synoptic-scale fronts, topographically-forced flows, or coastal convergence, as demonstrated in Section 4. In some cases, the sea breeze may interact with these other types of objects (Miller et al., 2003), and it may not be possible or meaningful to separate them. However, in other cases, the generality of the methods tested here may lead to misclassifications, especially in regions with steep topography or convectively active regions. We also do not consider the gravity wave aspect of the sea breeze that is produced by coastal diurnal heating (Rotunno, 1983), instead focusing on the thermally-direct circulation and surface-based front.

In addition, the ability of some diagnostics to identify the entire life-cycle of the sea breeze may be limited. For example, the hourly rate of change diagnostic (*H*, Coceal et al. (2018)), in the form applied in this study, appears to be biased towards the latter part of the sea breeze life cycle, when the front has already propagated more than 100 km onshore. This might be because the local change in conditions at inland locations are much sharper at later times of the day, combined with other sources of






local changes such as radiative cooling. The moisture frontogenesis (F) diagnostic may underestimate the true onshore extent of the sea breeze front, due to the dissipation of spatial gradients later in the life cycle. The sea breeze index (SBI, Hallgren et al. (2023)) does not consistently capture the onshore movement of the sea breeze, likely related to the circulation being most well-defined earlier in the life cycle near the coastline. Therefore, the methods demonstrated here might have more utility in determining whether a sea breeze may be present on a particular day or not, rather than capturing the entire life cycle and spatial extent.

There are also uncertainties here related to the filtering of objects and smoothing of convection-permitting model fields. Here, the same set of filters and smoothing parameters have been used for different regions and diagnostics (Table 2). For more accurate detections of sea breezes for individual coastlines, these filtering options could be tailored based on the application and diagnostic, including adjusting the spatial scales. We have demonstrated one set of filtering settings for general use around Australia, but this may remove some small-scale sea breeze objects such as on small tropical islands (see Section 4.2). In addition, we have not investigated the applicability of these methods to land breezes. It is not clear whether the *F* and *H* methods, which rely on gradients in moisture, will be useful over the ocean. Hallgren et al. (2023) proposed a land breeze index that characterizes the land breeze as a circulation in the same way as the *SBI*, although this was not explored in the current study.

We have applied sea breeze diagnostics to four models at different resolutions, ranging from 2.2 km to around 25 km horizontal grid spacing. The representation of sea breezes in the coarser-scale models (BARRA-R and ERA5) appears to vary depending on the region. For example, representations of the individual cases for Perth (Section 4.1) are very consistent between models, whereas the cases in tropical Australia vary between models considerably (Section 4.2). This is likely associated with different spatial scales of sea breezes in different regions, due to the scale of the coastline forcing. Statistics of object occurrences over the 6-month period also suggest that the higher resolution models (AUS2200 and BARRA-C) tend to produce a larger number of objects compared with ERA5 and BARRA-R. This is consistent with previous studies that have noted the need for km-scale models to properly resolve the sea breeze circulation (Cafaro et al., 2019).

Based on the findings presented here, we suggest that the moisture frontogenesis diagnostic (F) could be useful for capturing sea breeze occurrences from atmospheric model data, specifically when applied to convection-permitting model output (around 4 km or less). This is based on the utility of this diagnostic to identify the sea breezes cases in Section 4, while capturing a significant portion of the sea breeze life cycle with a maximum in object occurrences around the coastline and relatively few inland false detections (Section 5). A more practical benefit of applying the F diagnostic is that it requires near-surface winds and specific humidity only, in contrast to the SBI that requires model-level output. Model-level output is often unavailable and can require large amounts of computational resources to analyze. In addition, F can be applied to instantaneous time steps, in contrast to H that requires multiple time steps to calculate the rate of change in different variables. However, the limitations of the F diagnostic should be kept in mind in future applications, including the potential to misclassify other frontal objects and moist onshore large-scale winds flow as sea breezes. These limitations could potentially be avoided by further development of filtering methods.




These methods and findings have potential applications for climatological analyses of sea breeze occurrences across large regions, as well as facilitating the analysis of sea breeze occurrences for different regions using a single, robust method. Future work could apply this method for a climatological period over Australia to examine historical occurrences of sea breezes, including trends, the seasonal cycle, and the impact of different synoptic-scale conditions, while potentially quantifying the contribution of sea breezes to the diurnal cycle in wind speed. These sea breeze identification methods could also be used to help develop parameterisations of coastal rainfall in coarser models. Increased understanding and robust identification of sea breezes has implications for many coastal processes and human activity, for example, in human health and wind energy resource assessments.

Code and data availability. ERA5 data was used from the Australian NCI archive (https://dx.doi.org/10.25914/5fb115a9abecf) as well as the from the Analysis-Ready, Cloud Optimized data archive (https://console.cloud.google.com/marketplace/product/bigquery-public-data/arco-era5). The NCI also provided data for BARRA (https://dx.doi.org/10.25914/1x6g-2v48), AUS2200 (https://dx.doi.org/10.25914/w95d-q328), and Himawari satellite data processed by the Bureau of Meteorology (https://dx.doi.org/10.25914/61a609f9e7ffa). Bureau of Meteorology weather station data is available via their climate data service (http://www.bom.gov.au/climate/data/). Code for sea breeze object identification (sea\_breeze v1.1) is available here: https://doi.org/10.5281/zenodo.17220916 (Brown et al., 2025), while analysis code used to produce figures is available here: https://doi.org/10.5281/zenodo.17230239.

*Author contributions.* AB and CV conceptualized the research and designed the methods. AB and ES developed the software. AB, CV and ES contributed to writing. AB performed the formal analysis and visualization.

Competing interests. The authors declare that they have no conflict of interest.

Acknowledgements. This research was supported by the Australian Research Council Centre of Excellence for the Weather of the 21st Century (CE230100012), with the assistance of resources from the National Computational Infrastructure (NCI Australia), an NCRIS enabled capability supported by the Australian Government. We acknowledge the contributions of Jarrah Harrison-Lofthouse and in providing useful discussions on developing the methods, as well as Samuel Green for providing assistance with Python code optimization.

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
