# Peer review of "Identifying sea breezes from atmospheric model output (sea\_breeze v1.1)"

_EGUsphere, 2025_

## Author Comment (AC1)

**Response to review**

**Reviewer comment 1**

*https://doi.org/10.5194/egusphere-2025-4848-RC1*

The manuscript compares and contrasts three complementary methods of objectively identifying sea breezes in Australia, as applied to different analysis products of varying resolution. Model sensitivities are described and evaluated through physical argument, and satellite and surface weather station observations.

In general, the manuscript is well written with a clear experimental design and evaluation of the results. The authors also do a good job of critiquing the different methods and pointing out their shortcomings. The distance-dependent skill of different metrics (frontogenetic better near the coast and at early times; H less satisfactory in reproducing propagation) are interesting and helpful. Some of the results are not too surprising (analyses with resolution 10 km or finer are needed to properly capture these circulations, although even then apparently smoothing filters are needed due to model noise; topography and convection complicate results). Observational comparison is also limited, primarily based on satellite imagery for a few case studies; station-based evaluation is given in the supplement, but only for overall period frequencies.

I would recommend accepting the manuscript with only minor revisions as described below

**Specific Comments:**

- Line 18: I think 'onshore' generally means from sea to land. So I would rephrase this sentence to '...thermally-direct circulation with onshore flow and ascent over the land with descent over the sea.'
  Thanks for picking up on this sentence, as it was not written as intended. We meant to say that the sea breeze can be characterized as a: "...thermally-direct circulation with onshore surface flow and ascent over the land and offshore flow aloft with descent over the sea". This was intended to describe the sea breeze circulation that is relevant for certain diagnostics examined later in the manuscript. We have now changed this text to be as intended.

- Line 62: AUS2200 is a variant of the Australian Community Climate and Earth System Simulator -- for those not familiar with it, could you give more details on what it is, and what it means that it is a 'Simulator'?
  We have included more details on ACCESS in Section 2.2, with reference to a model description paper (Mackallah et al., 2022). We thought it was more appropriate to describe the model in the Data section, although we do reference this section in the Introduction when ACCESS is first mentioned. We have also tried to be clearer in describing how AUS2200 and BARRA relate to ACCESS. That

is, ACCESS is an Earth System Model, whereas AUS2200 and BARRA use only the atmospheric component of ACCESS in a regional configuration.

- Line 94: Why are specific humidity and air temperature given at 1.5 m instead of 2 m? Does AUS2200 have any surface fields, or only atmospheric level fields?
  The Unified Model outputs surface fields at a height of 1.5 m. We have clarified the text to state these are "surface" variables, to avoid potential confusion with atmospheric model level variables.

- Line 104: This seems to be the only place where you state what specifically the six-month period is. To assist the reader, maybe you could state the period (i.e., Jan-Feb 2013, 2016, 2018) at the end of the introduction?
  We have now stated the period in the second-last paragraph of the Introduction.

- Line 160: Maybe this is a digression from the main theme of the manuscript which is objective method sensitivity, but I am wondering why the only non-satellite based observational evaluation (the application of H to surface observations) is placed in the supplement, and also why it is used to only evaluate overall frequencies of events over the whole six-month period. I know observational verification of these phenomena is difficult, but could H be applied to the observations for at least the six selected case studies, to get a more fine-grained view of at least the accuracy of H?
  We appreciate this good point made by the reviewer. We have now included a comparison of *H* for each case between each model and station observations. This is shown in the Supplementary Material (Section S2.2, Figure S4) and discussed in Section 3.2. The new figure shows a spatial map of occurrences of high H values, with occurrences generally similar between each of the models and the observations.

  Given the length of the manuscript and number of figures already, we believe it is better to keep this analysis in the supplement, but we are happy to discuss this further.

- Line 196: State that the divergence here is the 2D divergence.
  Has been added, thank you

- Line 273: You say that you do not constrain the filter by time of day, but by using a filter that only allows land warmer than ocean, you are inherently only looking at sea breezes vs. land breezes, daytime vs. nighttime, and features over land vs. features over water, are you not?
  Yes, the reviewer is correct that the land-sea temperature filter will restrict objects to daytime occurrences and exclude land breezes. We have removed the text "we have chosen not to constrain the objects based on the time of day", and have instead explained that "the land-sea temperature filter restricts object occurrences to daytime and early evening hours, and excludes land breezes". Note that objects can still occur over water during the day, so long as the

neighbouring land is warmer than the air over the water.

- Line 357: Are there any references or data to generally support the geographical distribution of sea breeze events found in the analyses?
We are not aware of any studies that have analysed the spatial distribution of sea breezes in Australia, due to the limitations in identification methods mentioned in the Introduction. We believe that this is a major benefit for future work enabled by the methods presented in the manuscript, and have now mentioned this explicitly in the Discussion: "The methods presented here have enabled the assessment of the distribution of sea breeze across Australia for the first time, with opportunities for future work to investigate climatological patterns of occurrences. Climatological sea breeze occurrences based on these methods could also be compared with previous point-based climatologies in future work".

    In the submitted manuscript, we referenced a study where the distribution of sea breezes was analysed for small regions, in support of our findings (Clarke 1983, see Section 4.3 and Section 6). Similarly, we have now also included reference to Masselink and Pattiaratchi (2001) in Section 5.1.

- Supplement S1: The method to calculate coastline angle seems quite elaborate. But it appears to basically be a filter using length scales that are a function of the resolution of the analysis, at least at sufficient distances from the coastline. But the length scale of a natural phenomenon should be physically based. What would happen if R1 were chosen the same for all analyses? (Maybe ERA5 would be too noisy this way.)

    Rather than a filter based on length scales, the coastline angle method presented in Supplement S1 is intended to filter objects based on their orientation (relative to the coastline orientation), as well as their onshore wind speed (calculated using coastline orientation). We have tried to make this clearer in the revised manuscript (Section 3.1, first paragraph) as well as in the Supplementary Material (Section S1), with a clearer outline of the method.

    We do filter objects based on length scales using an area filter (see Table 2). This is done using pixel threshold rather than a consistent distance between models. As stated in the submitted manuscript, "This area criteria is applied on a pixel basis to account for larger circulations being resolved in coarser-scale models".

    To address the comment "What would happen if R1 were chosen the same for all analyses?", we have re-calculated the coastline angles from ERA5 using a value of 4 km for R1, compared to the original value of 50 km. This value of 4 km matches the value used for AUS2200. The figure below shows that this has some minimal impact on coastline angles near the coastline, but the distribution is broadly similar.

[Figure]

ERA5 coastline angles

- Supplement S3: I'm wondering if it would be possible to do sensitivity tests of this nature on H. Since it is a fuzzy logic algorithm, could you look at the impact of just including temporal jumps in just air temperature alone, or just humidity alone, for example?

  We have now performed this sensitivity test and have included the findings in Section S3.1 of the Supplementary Material. We performed three tests by removing moisture, temperature and onshore wind speed from the calculation of H. The results show that the diagnostic is most sensitive to the inclusion of moisture, with unrealistic inland propagation of objects if it is removed. The number of objects is also somewhat sensitive to the inclusion of temperature and moisture. However, the diurnal timing of objects is similar for all tests.

  We have also now noted these tests in the discussion of H (Section 3.2.3).

---

## Author Comment (AC2)

**Response to review**

**Reviewer comment 2**

*https://doi.org/10.5194/egusphere-2025-4848-RC2*

This work proposes three physically-based diagnostics for sea breeze events using atmospheric model outputs. The research topic is of interest to the research community. The manuscript is well structured and clearly written. I recommend a minor revision before being accepted for publication.

**Specific comments**

- There are resolution issues with all the figures.
  Apologies for the resolution issues. We created .jpeg figures with > 300 dpi, following the GMD manuscript submission guidelines, and embedded them in the pdf file, so we are not sure why there are resolution issues or how to fix this. We will mention this to the journal publication office.

- Ln 184: It might be helpful to be more specific about how the boundary layer height is determined. Did you use a fixed height, or did you post-calculated the PBLH?
  We have now noted that the PBLH is based on diagnostics output from the models, and have included some details on how these are calculated in the descriptions of model data (Sections 2.1 and 2.3)

- Ln 195: Add units for equation 3.
  We have now included more details on units in the text following Equation 3, for clarity.

- Did you find any systematic differences among the models and between models and obs, in sea breeze identification? Does high resolution model tend to diagnose sea breezes closer to obs?
  We have discussed this to some extent in the Discussion and Conclusion section of the submitted manuscript (7th paragraph of that section) and have now expanded on these points in the revised version in response to this comment. The key points are that higher resolution models (< 5 km grid) provide more detailed sea breeze fronts compared with coarser models (> 12 km grid), particularly for regions with complex coastlines, and produce more objects around Australia. In terms of being 'closer to obs', this is difficult to verify exactly due to a lack of a high-resolution observational network, and we mention this limitation now in the Discussion and Conclusion.

  We have also evaluated the models to a very limited extent using station observations applied to one of the sea breeze diagnostics (Supplement Section

S2), and this is mentioned (with added clarity) in Section 3.2 of the main text: "Because [the hourly rate of change diagnostic, H] is a time series method, it can also be applied to station observations and used as a model evaluation tool. This is explored in the Supplementary Material (Section S2.1), where it is shown that the models can represent spatial variations in observed hourly local changes over the entire 6-month period, as represented by the H diagnostic, with a high degree of accuracy. However, this depends on model resolution to some extent, with higher resolution models having higher correlations with observations".

- Do you predict any issues or have any recommendations for readers who would like to try your software and apply it to other regions?
  We believe that we have discussed a number of issues and recommendations in the Discussion and Conclusions of the submitted manuscript. For example:
    - L487: Based on the findings presented here, we suggest that the moisture frontogenesis diagnostic (F) could be useful for capturing sea breeze occurrences from atmospheric model data, specifically when applied to convection-permitting model output (around 4 km or less)
    - L491: A more practical benefit of applying the F diagnostic is that it requires near-surface winds and specific humidity only, in contrast to the SBI that requires model-level output
    - L494: The limitations of the F diagnostic should be kept in mind in future applications, including the potential to misclassify other frontal objects and moist onshore large-scale winds flow as sea breezes. These limitations could potentially be avoided by further development of filtering methods

  In the revised manuscript, we have now noted additionally in the Discussion and Conclusion that: "Care should be taken when applying the software and methods to regions of complex coastlines and topography, as well as convectively active regions, as these are potential sources of mis-classifications as shown in the results here".

  We have also amended the following paragraph to be more accurate in our recommendation:

  "Based on the findings presented here, we suggest that the moisture frontogenesis diagnostic (*F*) could be useful for capturing sea breeze occurrences from atmospheric model data across Australia using *sea_breeze*, specifically when applied to smoothed convection-permitting model output (around 4 km or less).... However, the limitations of the *F* diagnostic should be kept in mind in future applications to other regions or domains..."